# Demographic Control Measure Implications of Tuberculosis Infection for Migrant Workers across Taiwan Regions

**DOI:** 10.3390/ijerph19169899

**Published:** 2022-08-11

**Authors:** Szu-Chieh Chen, Tzu-Yun Wang, Hsin-Chieh Tsai, Chi-Yun Chen, Tien-Hsuan Lu, Yi-Jun Lin, Shu-Han You, Ying-Fei Yang, Chung-Min Liao

**Affiliations:** 1Department of Public Health, Chung Shan Medical University, Taichung 40201, Taiwan; 2Department of Family and Community Medicine, Chung Shan Medical University Hospital, Taichung 40201, Taiwan; 3Department of Bioenvironmental Systems Engineering, National Taiwan University, Taipei 10617, Taiwan; 4Department of Environmental Engineering, Da-Yeh University, Changhua 515006, Taiwan; 5Institute of Food Safety and Health Risk Assessment, National Yang Ming Chiao Tung University, Taipei 11221, Taiwan; 6Institute of Food Safety and Risk Management, National Taiwan Ocean University, Keelung City 20224, Taiwan

**Keywords:** tuberculosis, migrant worker, transmission dynamics, modeling, control measures

## Abstract

A sharp increase in migrant workers has raised concerns for TB epidemics, yet optimal TB control strategies remain unclear in Taiwan regions. This study assessed intervention efforts on reducing tuberculosis (TB) infection among migrant workers. We performed large-scale data analyses and used them to develop a control-based migrant worker-associated susceptible–latently infected–infectious–recovered (SLTR) model. We used the SLTR model to assess potential intervention strategies such as social distancing, early screening, and directly observed treatment, short-course (DOTS) for TB transmission among migrant workers and locals in three major hotspot cities from 2018 to 2023. We showed that social distancing was the best single strategy, while the best dual measure was social distancing coupled with early screening. However, the effectiveness of the triple strategy was marginally (1–3%) better than that of the dual measure. Our study provides a mechanistic framework to facilitate understanding of TB transmission dynamics between locals and migrant workers and to recommend better prevention strategies in anticipation of achieving WHO’s milestones by the next decade. Our work has implications for migrant worker-associated TB infection prevention on a global scale and provides a knowledge base for exploring how outcomes can be best implemented by alternative control measure approaches.

## 1. Introduction

Tuberculosis (TB) remains the leading cause of death from a single infectious agent and one of the top 10 causes of death worldwide; nearly 10 million people developed TB and 1.5 million died in 2020 [1]. There are 30 countries with high TB burdens, accounting for 86% of all estimated cases worldwide, of which eight of these countries account for two-thirds of the world’s TB cases: India (26%), China (8.5%), Indonesia (8.4%), the Philippines (6.0%), Pakistan (5.8%), Nigeria (4.6%), Bangladesh (3.6%), and South Africa (3.3%) [1].

Previous studies have suggested that migrant workers from countries with high TB incidences have a significant impact on TB epidemics in low-incidence countries [2,3,4,5,6,7,8,9]. Recently, the TB incidence in migrant workers (53–73.7 per 100,000 persons) was similar to that in the Taiwanese population (45.5–76.8 per 100,000 persons) in the period 2004–2013, whereas migrant workers in the youngest group (≤24 years old) had higher TB risk (5.3-fold) than the Taiwanese population [2]. In Taiwan, the average annual TB incidence rate was reported to be higher in the foreign-born population (94 per 100,000 persons) than in the local-born population (72 per 100,000 persons) in the period 2002–2005 [9]. The epidemiology of TB in foreign-born cases is predominant in females (65.4%), with greater incidence in the 25–44 year old population (70.9%), whereas the majority of cases among local-born population are male (69.4%) in the population aged >65 years (49.6%) [9]. The foreign-born population, regarded as a high-risk group, accounts for the majority of TB cases in Singapore, Finland, Japan, and Europe. Therefore, the sharp increase of 63% migrant workers from 280,928 in 2013 to 458,619 in 2019 has raised concerns for potential TB epidemics in Taiwan [10].

Nationwide epidemiological investigations have shown different risk factors among local and migrant groups [2]. It is evident that factors such as TB characteristics, population density, types of industries present in counties and cities, and variabilities between local and migrant workers can cause substantial differences in disease transmissions [2,8,9]. The substantial differences in TB incidence between Taiwanese and migrant workers might be partly due to high TB incidence after the arrival of migrant workers contributed by the reactivation of latent TB infections, as well as new TB infections in Taiwan [2,5]. Moreover, migrant workers experience more stress during their daily work and have less access to medical services, making them more vulnerable to infectious diseases [9].

The Taiwan Center of Disease Control (Taiwan CDC) has taken four strategies based on the signs and symptoms of TB to contain TB transmission in migrant workers: (i) active case finding by post-arrival medical examination and regular health examination, (ii) seeing a doctor when experiencing TB symptoms and culture-confirmed treatment, (iii) medical care through reimbursement policies, and (iv) implementing TB/multidrug-resistant TB (MDR-TB) case management through directly observed treatment, short-course (DOTS)/DOTS-plus programs [11]. Hospitals using case management with DOTS can improve the adherence of TB patients and the control of TB-epidemic situations [12]. The rates of acquired MDR-TB were significantly lower after the implementation of DOTS and DOTS-plus programs [13].

## 2. Background

A mechanistic, population transmission dynamics-based model can offer a useful tool for analyzing the spread of infectious diseases and evaluating the relative effectiveness of different measures [14]. The first mathematical model describing the epidemiological trend of TB was developed by Waaler et al. [15]. It consisted of five discrete difference equations, and model parameters were estimated from observed datasets. The first continuous TB model composed of ordinary differential equations was constructed by Revelle et al. [16]. Then, a simple three-compartment TB model was constructed by Blower et al. [17]. They developed it further into a five-compartment model to incorporate TB endogenous reactivation and relapse.

Optimal control theory is a branch of mathematics recently developed to obtain optimal methods to control a dynamic system [18,19,20]. The optimal control theory applied to TB models could provide valuable insight into the decision- and policymaking of public health agents. For example, Kim et al. [19] used a classic susceptible–latent–infected–recovery (SEIR) model to predict the effect of reducing the number of high-risk latent and infectious TB individuals and minimizing the cost of implementing control measures. These measures include reducing the contact between susceptible and infectious individuals, preventing the failure of treatment in infectious individuals, and increasing the treatment rate for target group such as people who have had close contact with TB patients and those who live with HIV patients.

Among various mathematical models of TB transmission, the factor of migration is rarely considered due to the assumption of closed populations [21]. We aimed to incorporate migrant workers as a subpopulation into a compartmental model to explore the intrinsic mechanism of TB transmission dynamics between migrant and local population. Several theoretical studies have formulated deterministic epidemiological models with immigration of infectives [22,23,24,25,26].

This study focused on examining the migrant-based TB transmission along with multiple control measures in highly endemic Taiwan regions. Hence, there were three aims of this study: (i) to establish an enhanced TB dynamic model based on Jia et al. [22] with an additional transmission route from infectious local to migrant populations, (ii) to assess the efforts of potential control strategies (social distancing, early screening, and DOTS) on TB incidence dynamics projected from the reference year 2018 to 2020 and 2023 further, after performing a broad literature data analysis to implement model parameterization, and (iii) to present different combinations of interventions for the optimal infection control. The percentage reduction in the populations of latently infected and infectious TB with/without control measures was the determinant modeling outcome.

## 3. Materials and Methods

### 3.1. Study Design

Concerning data availability and model implications on control measures, we adopted the model from Jia et al. [22] as our base model describing TB spread through horizontal transmission in the presence of exposed immigrants in host areas. The approach was the same with respect to considering the epidemiological and demographic factors that adequately explained regional TB incidence in the partitioned subpopulation: migrants and locals. In particular, we proposed a simple modification by considering the additional influence of the local population on the migrants, which was the limitation of Jia’s study [22]. Hence, we developed a migration-based susceptible–latently infected–infectious–recovered (SLTR) model to explore the TB mutual transmission between the infectious local population and migrant workers, as well as to evaluate how effectively various control measures are able to reduce infectives in the whole population (Figure 1).

### 3.2. Setting, Sample, and Data Collection

Here, we constructed a population−TB case dataset using the available visibility data in Taiwan regions based on a large-scale meta-analysis. Demographic data of nationality-based migrant workers in various productive industries in the period 2013–2019 were adopted from Taiwan Workforce Development Agency, Ministry of Labor (Appendix A) [27]. We showed that the three largest populations of migrant workers were located in Taoyuan City (91,898) and New Taipei City (56,041) of northern Taiwan and Taichung City (78,423) of central Taiwan (Appendix A). We, therefore, chose these three key administrative regions as our study areas. The TB incidence rates in the period 2012–2019 in Taiwan regions and source countries of migrant workers were adopted from data reported by the Taiwan CDC [28] and WHO [1], respectively. The yearly numbers of newly reported TB cases in the period 2006–2019 were also obtained from the Taiwan CDC [28].

### 3.3. Migrant-BASED SLTR Model

In the migrant-based SLTR model, the total population (*N*) comprised the migrant subpopulation (*N*_M_) and local subpopulation (*N*_L_), where *N* = *N*_M_ + *N*_L_. According to the natural history of TB, each subpopulation was further subdivided into four classes: susceptible (*S*), latently infected (*L*), those with infectious TB (*T*), and those that have recovered (*R*), resulting in eight groups of *S*_M_*, L*_M_*, T*_M_*, R*_M_, *S*_L_*, L*_L_*, T*_L_*,* and *R*_L_ (Figure 1A, Table 1).

Briefly, susceptible individuals can be infected by frequent contact with infectious TB cases (*T*_M_ and *T*_L_) and may progress from susceptible to latent TB. The parameter *Λ*_SM_ is the recruitment rate into *S*_M_ (persons·year^−1^), and *Λ*_SL_ is the crude birth rate into *S*_L_ (year^−1^). Newborn individuals are assumed susceptible (*Λ*_SL_). Parameters for arrivals of migrant workers include potentially susceptible (*Λ*_SM_), latently infected (*Λ*_LM_), or infectious TB (*Λ*_TM_), in that *β*_M_ and *β*_L_ are the transmission rates for migrant and local subpopulations (person^−1^·year^−1^), respectively, *β*_LM_ and *β*_ML_ are the transmission rates from locals to migrants and from migrants to locals (person^−1^·year^−1^), respectively, and *μ* is the background mortality rate (year^−1^).

Individuals with latent TB infection (LTBI) may develop active TB disease through a slow route, i.e., reactivation of TB in the remote future at rates *k*_M_ and *k*_L_, or a fast route, i.e., reinfection at rates *p*(*β*_M_*T*_M_ + *β_L_*_M_*T_L_*) and *p*(*β_L_T_L_* + *β_ML_T_M_*), in that *k*_M_ and *k*_L_ are the reactivation rates in *L*_M_ and *L*_L_ (year^−1^), respectively, *p* represents the previous infection affording partial immunity against reinfection, and *μ*_TM_ and *μ*_TL_ are the TB-induced mortality rates in *T*_M_ and *T*_L_ (year^−1^), respectively. The progression rates from infectious TB to recovery determined by *γ*_M_ and *γ*_L_ are the recovery rates of *T*_M_ and *T*_L_ (year^−1^), respectively. Additionally, *p* and *q* are the partial immunities that decrease the probability of progression to TB after reinfection for latent and recovered individuals, respectively.

### 3.4. Control-Based SLTR Model

Here, we considered three TB constant control strategies: social distancing control (*u*_1_), early screening (*u*_2_), and DOTS (*u*_3_). We incorporated the selected control measures into the SLTR model to form the control-based SLTR model (Figure 1B, Table 2). We used the present developed control-based SLTR model to assess the efforts of potential control strategies on TB incidence dynamics projected from the reference year 2018 to 2020 and 2023 further.

Briefly, the coefficient 1 − *u*_1_ represents the effort of reducing susceptible individuals that become infected by infectious individuals (*β*_L_, *β*_M_, *β*_ML_, and *β*_LM_), such as isolation of infectious people or health education for disease prevention and control. The coefficient 1 − *u*_2_ represents the effort of reducing the risk of progression to the infectious state (*k*_L_ and *k*_M_), through measures such as identification and screening of latent individuals who are at high risk of developing TB. Notably, two controls *u*_1_ and *u*_2_ are bound within 10–90%. In light of DOTS, 1 + *u*_3_ represents the effort of increasing *γ*_M_ and *γ*_L_, such as improving medication adherence in patients, which is essential at least during the intensive phase of treatment (the first 2 months) to ensure that the drugs are taken in the right combinations and for the appropriate duration. The DOTS strategy can be used to treat the infectious TB cases. Thus, the completely cured TB patients will shift into the recovery/treated classes. Based on the Taiwan CDC report [29], the recovery rate for treatment cases has reached 70%; hence, we propose a simulation of much higher medical care with the interval ranging from 80–95%. To investigate the impact or effectiveness of the control strategies, different control intervention schemes were considered in single (*u*_1_, *u*_2_, and *u*_3_), dual (*u*_1_ + *u*_2_, *u*_1_ + *u*_3_, and *u*_2_ + *u*_3_), and triple (*u*_1_ + *u*_2_ + *u*_3_) combinations (Figure 1C). Therefore, the percentage reduction in the populations of latently infected *L* (*L*_L_
*+ L*_M_) and infectious TB *T* (*T*_L_
*+ T*_M_) as the modeling outcomes could be estimated.

### 3.5. Model Parameterization

Initial population sizes and parameters used in the migrant-based SLTR model involving Taoyuan City, Taichung City, and New Taipei City were estimated on the basis of the available site–country-specific visibility TB data in the reference year of 2018 provided by the Taiwan CDC and the published literature.

### 3.6. Data and Sensitivity Analyses

To determine the relative importance of model parameters against the TB transmission and population dynamics, we performed a sensitivity analysis against five parameters known as the transmission rate (*β*_L_, *β*_M_, *β*_ML_, and *β*_LM_) and reactivation rate (*k*) based on the previous studies [18,19,20]. We set the parameter intervals of *β*_L_ and *β*_ML_ at 5 × 10^−5^–5 × 10^−9^ person^−1^·year^−1^, while *β*_M_ and *β*_LM_ were set at 5.9172 × 10^−5^–5.9172 × 10^−9^ and 10^−5^–10^−9^ person^−1^·year^−1^, respectively [30]. On the other hand, parameter *k* was set to 0.0007 [17], 0.0024 [30], 0.004 [31], 0.0294 [19], and 0.0527 [22]. Model simulations were performed using Berkeley Madonna 8.39 (Berkeley Madonna was developed by Robert Macey and George Oster of the University of California, Berkeley, CA, USA).

## 4. Results

### 4.1. Data Interpretations

Nationality-based migrant workers from Indonesia, the Philippines, and Vietnam substantially increased from 45,919, 67,442, and 104,590 in 2013 to 74,764, 126,661, and 196,162, respectively, in 2019 (Figure 2A). This accounts for increases of 63% (Indonesia), 88% (the Philippines), and 88% (Vietnam) of foreign-born workers during the 6 years, implying that the largest two groups of migrant workers were consistently from Vietnam and the Philippines. As a matter of fact, Vietnam and the Philippines accounted for 43% and 28% (16% for Indonesia; 13% for Thailand) of the total numbers of migrant workers in Taiwan in 2019. Generally, the nationality-based migrant workers as percentages of the total numbers in each city and county varied substantially varied on the basis of a geographic distribution with a median of 7200 (2.5th–97.5th percentile: 60–91,900) (Figure 2B,C; Appendix A).

We found that the incidence rate in the Philippines was 554 per 100,000 population in 2019, which was much higher than that in Indonesia (312 per 100,000 population), Vietnam (176 per 100,000 population), Thailand (150 per 100,000 population), and Taiwan (37 per 100,000 population) (Figure 3A). Our results indicated a substantial increase in TB cases among migrant workers from 2006 to 2014, while the confirmed cases (600–700) have remained steady since 2014 (Figure 3B). Furthermore, of the total 6416 (2006–2019 summation) migrant TB cases, Indonesia was the largest contributor (47%), followed by the Philippines (25%), Vietnam (18%), Thailand (9%), and others (1%: 38 cases from Malaysia, Japan, Korea, and unknown nationalities) (Figure 3C).

### 4.2. Sensitivity Performance

An overall presentation of all sensitivity tests against the four transmission rates (*β*_M_, *β*_L_, *β*_LM_, and *β*_ML_) and reactivation rate (*k*) for Taoyuan City, Taichung City, and New Taipei City is available in Appendix A. Our results showed that the dynamic modeling with transmission rates for the migrant population (*β*_M_ = ~6 × 10^−5^) and transmission rates for the local subpopulation (*β*_L_ = 5 × 10^−5^) were crucial compared to other estimates. The migrant subpopulation (*T*_M_) and local subpopulation (*T*_L_) will approximately reach a maximum population by 2022–2023 in Taoyuan City. When the reactivation rate (*k*) was increased, both migrant subpopulation (*T*_M_) and local subpopulation (*T*_L_) slowly grew among three study cities, but there were no significant effects on *N*_M_, *S*_M_, *N*_L_, or *S*_L_ (Appendix A).

### 4.3. Single Control Effects

The estimates of initial population sizes in the reference year of 2018 in the three selected TB hotspot cities applied in the control-based SLTR model are listed in Table 3. Results of model parameterizations in the SLTR model are shown in Table 4. Our simulation results showed that, in terms of reducing the total numbers of *L* + *T*, distancing (*u*_1_) was the most efficient intervention when a single control strategy was used, followed by early screening (*u*_2_) and DOTS (*u*_3_) (Figure 4, Table 5, and Appendix A).

In the 2 year projection in Taoyuan City (Table 5), *u*_1_ = 10% could reduce *L* + *T* by ~6% more. When *u*_1_ effort increased to 90%, ~42% of *L* + *T* could be reduced. On the other hand, in the 5 year projection, *L* + *T* could be reduced by ~18% with *u*_1_ = 10%. When *u*_1_ increased to 40% and 90%, *L* + *T* could be reduced by ~50% and ~80%, respectively (Figure 4A). In Taichung City, *L* + *T* could be reduced by ~52% with *u*_1_ = 30% control (Figure 4A). When *u*_1_ effort increased to 90%, there was a reduction of ~85% in *L* + *T* (projecting five years) (Table 5). Similarly, in New Taipei City, *L* + *T* could be reduced by ~53% and ~97% with *u*_1_ = 10% and 90% efforts, respectively (projecting five years) (Table 5).

Taken together, in all single control strategies (Figure 4A–C), we showed that a higher control effort led to a more pronounced reduction in the *L* + *T* group, and a longer time control resulted in a better effort. Therefore, if only one control strategy is to be selected, distancing (*u*_1_) would be the most effective one.

### 4.4. Dual and Triple Combination Control Strategies Effects

In this study, dual and triple combination control strategies (*u*_1_ + *u*_2_, *u*_1_ + *u*_3_, *u*_2_ + *u*_3_ and *u*_1_ + *u*_2_ + *u*_3_) were also simulated (Figure 4D, Appendix A). Here, we used the efforts of *u*_1_ = 70%, *u*_2_ = 70%, and *u*_3_ = 85% to simplify our results. In a 2 year projection in Taoyuan City (Figure 4D), *L* + *T* could be reduced by ~39% through *u*_1_ + *u*_2_ control. In the 5 year simulation, *u*_1_ + *u*_2_ effort could reduce *L* + *T* by ~74% in Taoyuan City, ~83% in Taichung City, and ~96% in New Taipei City (Figure 4D). Moreover, the dual control *u*_1_ + *u*_2_ had the same effort as *u*_1_ + *u*_3_. In New Taipei City (Figure 4D), distancing (*u*_1_) coupled with early screening (*u*_2_) or DOTS (*u*_3_) was the most effective combination in curtailing *L* + *T*. Furthermore, the *u*_1_ + *u*_2_ + *u*_3_ combination could reduce *L* + *T* by ~97% in a 5 year projection (Figure 4D).

## 5. Discussion

This study used the city-specific demographic characteristics in the control-based SLTR model with identity transmission rates (the most sensitive parameter) in the three cities due to the lack of available data. Despite this limitation, the present study demonstrated that, among the single control strategies, distancing control (*u*_1_) was the most effective in curtailing cases of latent and active TB (*L* + *T*), consistent with previous findings [18,19]. Our simulations showed that implementing distancing control would reduce *L* + *T* by ~35–57% in 2 years and ~70–95% in 5 years. Not surprisingly, a combination of control strategies would yield more additional reductions. Distancing coupled with latent case finding (*u*_1_ + *u*_2_) was the most effective combination, further reducing the *L* + *T* by ~5%. Since this coupled control implementation would continuously decrease *L* + *T*, it is expected to attain the WHO’s 2025 milestone (50% reduction in incidence rate by the year 2025) and 2035 target (90% reduction in the incidence rate by the year 2035).

Pre-entry screening programs for TB in migrant workers are believed to be a high-yield policy targeting active TB [5]. According to Taiwan’s policy [36,37], newly arrived migrant workers from highly endemic countries are required to have a verified normal chest X-ray (CXR) performed overseas 3 months prior to entry and at various times post entry, i.e., post-entry screenings at 0–3 days (first round), 6 months (second round), 18 months (third round), and 30 months (fourth round) after arrival to obtain residency for a maximum of 3 years [38]. When employed migrants are confirmed by health examination to have active TB, TB pleurisy, or Hansen’s disease, except for cases of multiple drug resistance, the employer should submit documents to the municipality or county (city) competent health authorities to apply for DOTS services within 15 days from the next day after receiving the diagnosis certificate [37]. However, the current screening tool and chest radiography are sensitive to detect active pulmonary TB but insufficient to catch LTBI [2]. Hence, improved diagnosis and effective treatment for LTBI among newly arrived individuals at risk are a major focus for the control and elimination of TB in the United States and other low-incidence countries [39,40,41,42].

Countries with a high burden of MDR-TB such as the Philippines, Indonesia, Vietnam, and Thailand are neighboring countries with close and frequent contacts with the Taiwanese population, and they represent the main importing countries of Taiwan’s migrant workers. In recent years, China and Vietnam have imported the most MDR-TB cases from abroad (or have a history of going abroad), accounting for about 13% of reported MDR-TB cases [43]. Statistics show that the number of foreigners diagnosed with TB has increased from ~450 in 2008 and remained at ~850 in the past 5 years [43]. There are no relevant data to assist in estimating the transmission rate, reinfection rate, and other parameters among different regions, ages, and genders. To conduct model validations on the basis of the current available data, we compared the actual TB cases with our modeling outcomes and observed a trend of underestimations in model simulations in 2019 and 2020. However, the trend of decreasing case numbers from 2019 to 2020 in model simulation was in accordance with the changes in real case numbers in the same period [11]. We, thus, inferred that the lower estimates of the infected population in the three target cities could be ascribed to the lower reactivation rate in the latent population. We constructed a conservative model differentiating latent and infected TB cases into two different populations of the targeted three cities with most migrants in Taiwan. However, reactivation rates have been found to have high discrepancies in many studies [31,44,45]. We adopted the most conservative reactivation rate estimate (0.004) for the model simulations. However, the annual trend of TB case numbers could be predicted. The established model and derivations of the epidemiological parameters are also adaptable to different countries or regions with the issue of TB-infected migrant workers. Hence, with applications of the current constructed mathematical models, we considered a completely mixed and homogeneous system to simplify the complicated infection process. Furthermore, this study only assumed that the effectiveness of the control strategy reached a certain value. We did not consider the cost-effectiveness analysis associated with the type of control strategy since there is a lack of relevant data for calculating cost-related weights in Taiwan.

## 6. Conclusions

We used a parsimonious control-based SLTR transmission model to account for the mutual TB transmission between local populations and migrant workers for assessing the impact of potential control strategies on TB infection in three major hotspot cities in Taiwan regions, and we provided suggestions for sound control strategies that should be implemented. Overall, our findings suggest that the social distancing is the best single control strategy, while the best dual control is social distancing together with early screening. However, the effectiveness of the triple control strategy was marginally (1–3%) better than the dual measure. Therefore, if there are sufficient resources, a dual strategy should be implemented, which will reduce the total numbers of latently infected and infectious individuals more effectively than a single control, while also ensuring lower costs than the triple strategy. Our work has implications for migrant worker-associated TB infection prevention on a global scale and provides a knowledge base for exploring how outcomes can be best implemented by alternative control measure approaches.

## Figures and Tables

**Figure 1 ijerph-19-09899-f001:**
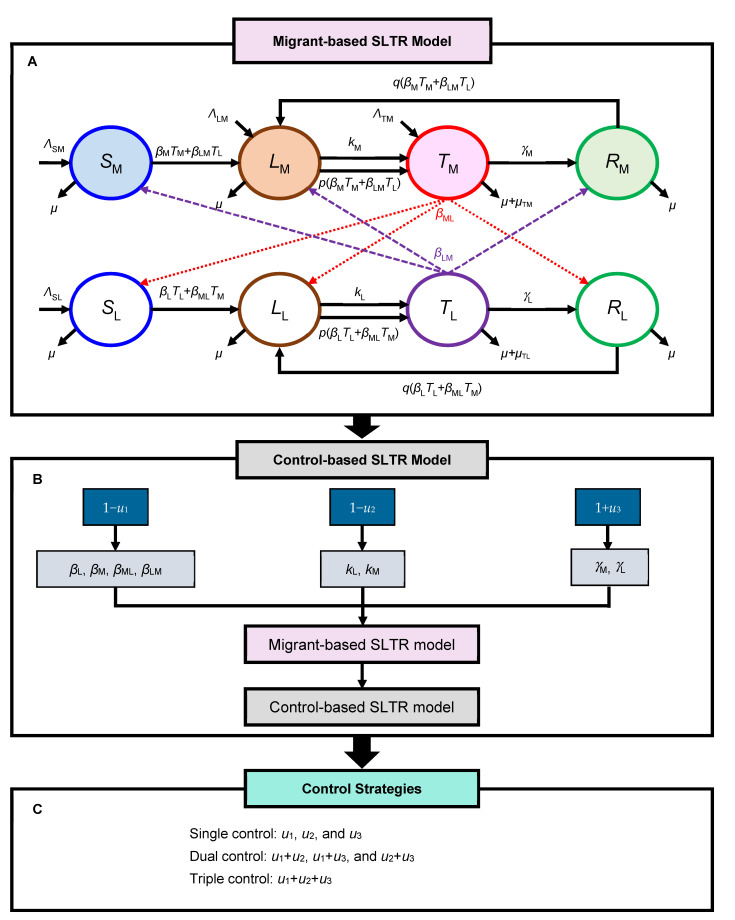
Schematic showing (**A**) the migration-based TB transmission model. The compartmental susceptible (*S*)–latently infected (*L*)–infectious tuberculosis (*T*)–recovered (*R*) (SLTR) model for assessing the impact of migration on TB epidemics on a regional scale. *M*: migrant subpopulation; *L*: local subpopulation. (**B**) The control-based SLTR model with different control strategies (combinations) based on distancing control (*u*_1_), early screening control (*u*_2_), and directly observed treatment, short-course (DOTS) control (*u*_3_). (**C**) The control intervention schemes were considered in single, dual, and triple combinations (see text for symbol meanings).

**Figure 2 ijerph-19-09899-f002:**
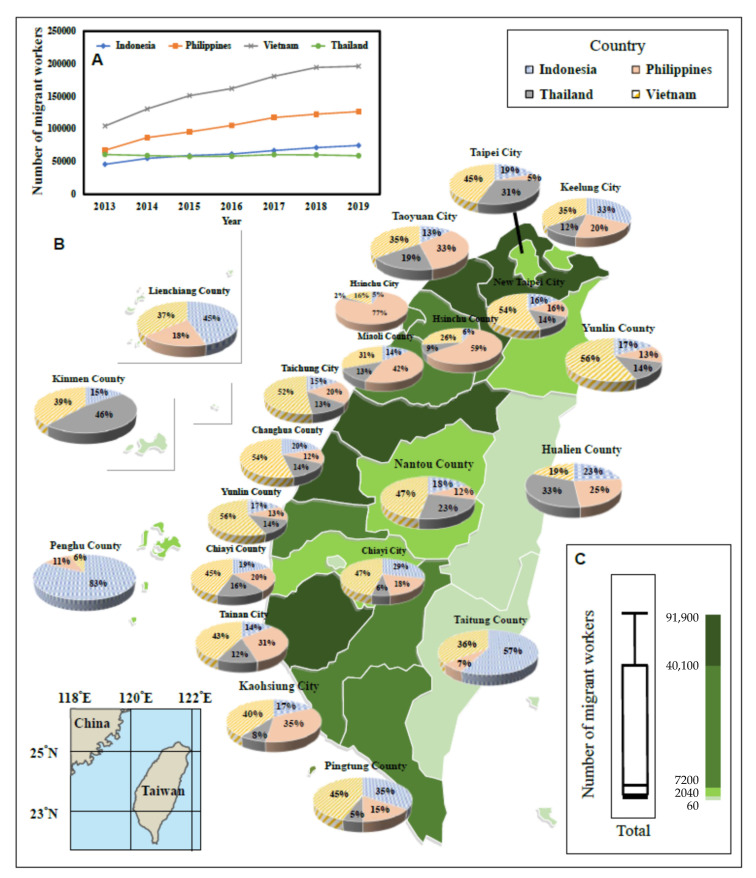
(**A**) Numbers of migrant workers in Taiwan from Indonesia, the Philippines, Thailand, and Vietnam per year in the period 2013–2019. (**B**) Migrant workers by nationality as percentages of the total numbers in each city and county in 2018. (**C**) Statistical distribution of total number of migrant workers according to box–whisker plot (box plot: 25th–75th percentile; whisker plot: 2.5th–97.5th percentile; middle line: median). Data were adopted from the Ministry of Labor Republic [27].

**Figure 3 ijerph-19-09899-f003:**
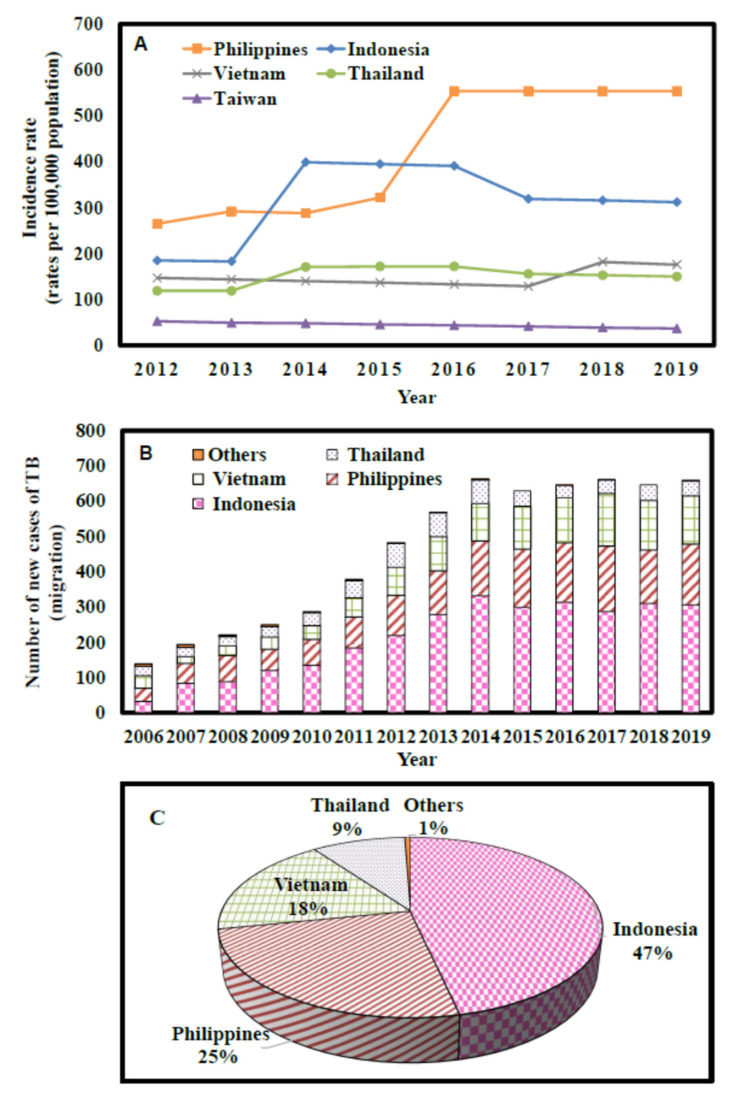
(**A**) Country-specific incidence rates per 100,000 population in the period 2012–2019 in Taiwan region (WHO [1]). (**B**) Cases of confirmed tuberculosis (TB) among migrant workers in Taiwan in the period 2006–2019 (Taiwan CDC [28]). (**C**) Country-specific contribution percentage (%) among the confirmed cases of TB (Taiwan CDC [28]).

**Figure 4 ijerph-19-09899-f004:**
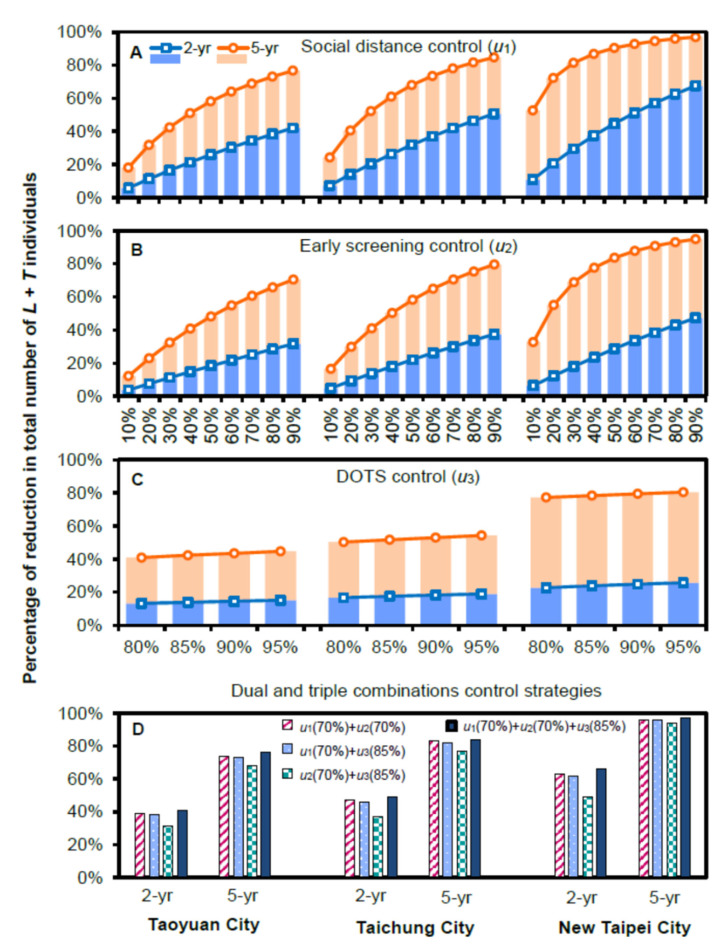
Percentage reduction in total number of latently infected (*L*) + infectious (*T*) individuals under single control of (**A**) social distancing control (*u*_1_), (**B**) early screening control (*u*_2_), and (**C**) DOTS control (*u*_3_), or (**D**) dual (*u*_1_ + *u*_2_, *u*_1_ + *u*_3_, and *u*_2_ + *u*_3_) and triple (*u*_1_ + *u*_2_ + *u*_3_) combinations control strategies in Taoyuan, Taichung, and New Taipei Cities in 2020 (2 year projection) and 2023 (5 year projection).

**Table 1 ijerph-19-09899-t001:** System equations used in the migrant-based SLTR model (see text for symbol meanings).

Symbol	Equation	
Migrant population (*N*_M_)		
*S* _M_	$\frac{dS_{M}}{dt}=\Lambda_{\mathrm{SM}}-(\beta_{M}T_{M}+\beta_{LM}T_{L})S_{M}-\mu S_{M}$	(1)
*L* _M_	$\frac{dL_{M}}{dt}=\Lambda_{\mathrm{LM}}+(\beta_{M}T_{M}+\beta_{LM}T_{L})S_{M}+q(\beta_{M}T_{M}+\beta_{LM}T_{L})R_{M}-k_{M}L_{M}-p(\beta_{M}T_{M}+\beta_{LM}T_{L})L_{M}-\mu L_{M}$	(2)
*T* _M_	$\frac{dT_{M}}{dt}=\Lambda_{\mathrm{TM}}+k_{M}L_{M}+p\left(\beta_{M}T_{M}+\beta_{\mathrm{LM}}T_{L}\right)L_{M}-\gamma_{M}T_{M}-\left(\mu +\mu_{\mathrm{TM}}\right)T_{M}$	(3)
*R* _M_	$\frac{dR_{M}}{dt}=\gamma_{M}T_{M}-q\left(\beta_{M}T_{M}+\beta_{\mathrm{LM}}T_{L}\right)R_{M}-\mu R_{M}$	(4)
Local population (*N*_L_)		
*S* _L_	$\frac{dS_{L}}{dt}=\Lambda_{SL}-(\beta_{L}T_{L}+\beta_{ML}T_{M})S_{L}-\mu S_{L}$	(5)
*L* _L_	$\frac{dL_{L}}{dt}=(\beta_{L}T_{L}+\beta_{ML}T_{M})S_{L}+q(\beta_{L}T_{L}+\beta_{ML}T_{M})R_{L}-k_{L}L_{L}-p(\beta_{L}T_{L}+\beta_{ML}T_{M})L_{L}-\mu L_{L}$	(6)
*T* _L_	$\frac{dT_{L}}{dt}=k_{L}L_{L}+p\left(\beta_{L}T_{L}+\beta_{\mathrm{ML}}T_{M}\right)L_{L}-\gamma_{L}T_{L}-\left(\mu +\mu_{\mathrm{TL}}\right)T_{L}$	(7)
*R* _L_	$\frac{dR_{L}}{dt}=\gamma_{L}T_{L}-q\left(\beta_{L}T_{L}+\beta_{ML}T_{M}\right)R_{L}-\mu R_{L}$	(8)

**Table 2 ijerph-19-09899-t002:** System equations used in the control-based SLTR model based on three control measures *u*_1_, *u*_2_, and *u*_3_ denoting the efforts of social distancing control, early screening control, and directly observed treatment, short-course (DOTS) control, respectively (see text for symbol meanings).

Migrant Population (*N*_M_)	
$\frac{dS_{M}}{dt}=\Lambda_{\mathrm{SM}}-\left(\left(1-u_{1}\right)\beta_{M}T_{M}+\left(1-u_{1}\right)\beta_{\mathrm{LM}}T_{L}\right)S_{M}-\mu S_{M}$	(9)
$\frac{dL_{M}}{dt}=\Lambda_{\mathrm{LM}}+\left(\left(1-u_{1}\right)\beta_{M}T_{M}+\left(1-u_{1}\right)\beta_{\mathrm{LM}}T_{L}\right)S_{M}+q\left(\left(1-u_{1}\right)\beta_{M}T_{M}+\left(1-u_{1}\right)\beta_{\mathrm{LM}}T_{L}\right)R_{M}-\left(1-u_{2}\right)k_{M}L_{M}-p\left(\left(1-u_{1}\right)\beta_{M}T_{M}+\left(1-u_{1}\right)\beta_{\mathrm{LM}}T_{L}\right)L_{M}-\mu L_{M}$	(10)
$\frac{dT_{M}}{dt}=\Lambda_{\mathrm{TM}}+\left(1-u_{2}\right)k_{M}L_{M}+p\left(\left(1-u_{1}\right)\beta_{M}T_{M}+\left(1-u_{1}\right)\beta_{\mathrm{LM}}T_{L}\right)L_{M}-\left(1+u_{3}\right)\gamma_{M}T_{M}-\left(\mu +\mu_{\mathrm{TM}}\right)T_{M}$	(11)
$\frac{dR_{M}}{dt}=\left(1+u_{3}\right)\gamma_{M}T_{M}-q\left(\left(1-u_{1}\right)\beta_{M}T_{M}+\left(1-u_{1}\right)\beta_{\mathrm{LM}}T_{L}\right)R_{M}-\mu R_{M}$	(12)
*N*_M_ = *S*_M_ + *L*_M_ + *T*_M_ + *R*_M_	(13)
Local population (*N*_L_)	
$\frac{dS_{L}}{dt}=\Lambda_{\mathrm{SL}}-\left(\left(1-u_{1}\right)\beta_{L}T_{L}+\left(1-u_{1}\right)\beta_{\mathrm{ML}}T_{M}\right)S_{L}-\mu S_{L}$	(14)
$\frac{dL_{L}}{dt}=\left(\left(1-u_{1}\right)\beta_{L}T_{L}+\left(1-u_{1}\right)\beta_{\mathrm{ML}}T_{M}\right)S_{L}+q\left(\left(1-u_{1}\right)\beta_{L}T_{L}+\left(1-u_{1}\right)\beta_{\mathrm{ML}}T_{M}\right)R_{L}-\left(1-u_{2}\right)k_{L}L_{L}-p\left(\left(1-u_{1}\right)\beta_{L}T_{L}+\left(1-u_{1}\right)\beta_{\mathrm{ML}}T_{M}\right)L_{L}-\mu L_{L}$	(15)
$\frac{dT_{L}}{dt}=\left(1-u_{2}\right)k_{L}L_{L}+p\left(\left(1-u_{1}\right)\beta_{L}T_{L}+\left(1-u_{1}\right)\beta_{\mathrm{ML}}T_{M}\right)L_{L}-\left(1+u_{3}\right)\gamma_{L}T_{L}-\left(\mu +\mu_{\mathrm{TL}}\right)T_{L}$	(16)
$\frac{dR_{L}}{dt}=\left(1+u_{3}\right)\gamma_{L}T_{L}-q\left(\left(1-u_{1}\right)\beta_{L}T_{L}+\left(1-u_{1}\right)\beta_{\mathrm{ML}}T_{M}\right)R_{L}-\mu R_{L}$	(17)
*N*_L_ = *S*_L_ + *L*_L_ + *T*_L_ + *R*_L_	(18)

**Table 3 ijerph-19-09899-t003:** Estimation of the initial population sizes (persons) in the reference year of 2018 in three selected TB hotspot cities applied in the control-based SLTR model.

Symbol	Description	Estimate		
		Taoyuan City	Taichung City	New Taipei City
Local subpopulation				
*S*_L_(0) ^d^	Susceptible individuals in the local subpopulation	2,211,473	2,791,803	3,978,357
*L*_L_(0) ^c^	Latently infected individuals in the local subpopulation	8173	10,318	14,704
*T*_L_(0) ^a^	Infectious TB cases in the local subpopulation	662	1007	1509
*R*_L_(0) ^b^	Recovered cases in the local subpopulation	484	766	1077
Migrant subpopulation				
*S*_M_(0) ^g^	Susceptible individuals in the migrant subpopulation	63,461	54,156	38,699
*L*_M_(0) ^f^	Latently infected individuals in the migrant subpopulation	28,305	24,154	17,261
*T*_M_(0) ^e^	Infectious TB cases in the migrant subpopulation	133	113	81
*R*_M_(0)	Recovered cases in the migrant subpopulation	0	0	0

^a^ Adopted from [11]. ^b^ Estimated by multiplying *T*_L_(0) with cured rate in a specific city: 73.1% (Taoyuan City), 76.1% (Taichung City), and 71.4% (New Taipei City) [11]. ^c^ *L*_L_(0) = 0.004 × (1 − 0.08) × *N*_L_(0) = 8173, 10,318, and 14,704, respectively, where 0.004 is the annual infection risk [32], and 0.08 is the probability of new infections that develop progressive primary active TB [33] with total local subpopulations. *N*_L_(0) = 2,220,792 (Taoyuan City), 2,803,894 (Taichung City), and 3,995,647 (New Taipei City). ^d^
*S*_L_(0) = *N*_L_(0) − *L*_L_(0) − *T*_L_(0) − *R*_L_(0). ^e^
*T*_M_(0) was estimated as new TB cases of migrant workers in 2018 in Taiwanese population × (*N*_Tao_, _Tai_, _NT_ / *N*). *N*_Tao_, *N*_Tai_, and *N*_NT_ indicate the number of migrant workers in Taoyuan City, Taichung City, and New Taipei City, respectively, whereas *N* is the total number of migrant workers in Taiwan. ^f^
*L*_M_(0) is estimated as *N*_Tao,Tai,NT_ × latent TB prevalence from WHO Southeast Asia data as 30.8% (95% CI: 28.3–34.8%) adopted from [34]. ^g^
*S*_M_(0) = *N*_M_(0) − *L*_M_(0) − *T*_M_(0) − *R*_M_(0), where *N*_M_(0) is estimated according to [35].

**Table 4 ijerph-19-09899-t004:** Parameter estimation and reference information used in the control-based SLTR models among Taoyuan City, Taichung City, and New Taipei City.

Symbol	Description	Taoyuan City	Taichung City	New Taipei City
*Λ* _SM_	Recruitment rate into *S*_M_ (person·year^−1^)	24,394	20,818	14,876
*Λ* _LM_	Recruitment rate into *L*_M_ (person·year^−1^)	10,864	9271	6625
*Λ* _TM_	Recruitment rate into *T*_M_ (person·year^−1^)	15	12	9
*Λ* _SL_	Crude birth rate into *S*_L_ (year^−1^)	0.0102	0.0081	0.0072
*β*_L_ ^a^	Transmission rate for the local subpopulation (person^−1^·year^−1^)	5 × 10^−7^	5 × 10^−7^	5 × 10^−7^
*β*_M_ ^a^	Transmission rate for the migrant subpopulation (person^−1^·year^−1^)	5.9172 × 10^−7^	5.9172 × 10^−7^	5.9172 × 10^−7^
*β*_ML_ ^a^	Transmission rate for migrants in the local subpopulation (person^−1^·year^−1^)	5 × 10^−9^	5 × 10^−9^	5 × 10^−9^
*β*_LM_ ^a^	Transmission rate for locals in the migrant subpopulation (person^−1^·year^−1^)	10^−8^	10^−8^	10^−8^
*k*_M_ ^b^	Reactivation rate in *L*_M_ (year^−1^)	0.004	0.004	0.004
*k*_L_ ^b^	Reactivation rate in *L*_L_ (year^−1^)	0.004	0.004	0.004
$\gamma_{M}$ ^c^	Recovery rate of *T*_M_ (year^−1^)	0.731	0.761	0.714
$\gamma_{L}$ ^c^	Recovery rate of *T*_L_ (year^−1^)	0.731	0.761	0.714
*µ*	Background mortality rate (year^−1^)	0.0578	0.0611	0.06
$\mu_{\mathrm{TM}}$ ^c^	TB-induced mortality rate in *T*_M_ (year^−1^)	0.187	0.174	0.195
$\mu_{\mathrm{TL}}$ ^c^	TB-induced mortality rate in *T*_L_ (year^−1^)	0.187	0.174	0.195
*p* ^d^	Partial immunity that decreases the probability of fast progression after reinfection for *T*_L_	0.8	0.8	0.8
*q* ^d^	Partial immunity that decreases the probability of fast progression after reinfection for *R*_L_	0.8	0.8	0.8

Note: *Λ*_SM_ is estimated as the number of migrant workers who took the entry examination from 2018 × (*N*_Tao,Tai_, _NT_/*N*) − *Λ*_LM,*I*_ − *Λ*_TM,*i*_, adopted from [10,28]. *Λ*_LM_ is estimated as the number of migrant workers who took entry examination from 2018 × (*N*_Tao,Tai_, _NT_/*N*) × latent TB prevalence in WHO Southeast Asia. *Λ*_TM_ is estimated as the number of migrant workers who failed the TB examination via chest X-ray at the health examination within 3 days of arrival from 2018 × (*N*_Tao,Tai_, _NT_/*N*). *Λ*_SL_ (year^−1^). Background mortality rates *μ* in Taoyuan City, Taichung City and New Taipei City were estimated to be 0.0102 and 0.0578, 0.0081 and 0.0611, and 0.0072 and 0.06, respectively, cited from the Department of Statistics, Ministry of the Interior [35]. ^a^ Adopted from [30]. ^b^ Adopted from [31]. ^c^ Adopted from [11]. ^d^ Adopted from [22].

**Table 5 ijerph-19-09899-t005:** The corresponding effort on the total number of latently infected *L* (*L_L_ + L_M_*) and infectious individuals *T* (*T_L_ + T_M_*) under different control strategies in the period 2018–2020 and projection to 2022–2023 in Taoyuan City, Taichung City, and New Taipei City. *u*_1_: social distancing control, *u*_2_: early screening, and *u*_3_: DOTS.

Total Number of *L* + *T* (Individuals)							
Year	Without Control	*u*_1_ = 10%	*u*_1_ = 90%	*u*_2_ = 10%	*u*_2_ = 90%	*u*_3_ = 80%	*u*_3_ = 95%
Taoyuan City							
2018	37,295	37,295	37,295	37,295	37,295	37,295	37,295
2019	59,767	58,104	45,999	58,879	51,913	56,642	56,159
2020	93,665	88,229	54,258	90,079	63,927	81,213	79,516
2021	142,652	129,680	62,012	133,700	75,017	111,404	107,606
2022	214,558	186,689	69,239	195,390	85,766	148,114	140,984
2023	326,190	267,077	75,937	286,527	96,386	192,655	180,420
Taichung City							
2018	35,515	35,515	35,515	35,515	35,515	35,515	35,515
2019	61,642	59,356	43,002	60,537	51,948	57,303	56,640
2020	101,770	94,333	50,153	97,070	63,911	84,759	82,490
2021	163,894	145,302	56,883	151,335	74,338	120,001	114,859
2022	264,941	221,243	63,156	235,232	84,208	165,267	155,276
2023	450,456	341,343	68,956	377,625	93,855	223,944	205,858
New Taipei City							
2018	33,575	33,575	33,575	33,575	33,575	33,575	33,575
2019	73,941	69,572	39,767	72,206	58,982	66,268	65,089
2020	140,817	125,403	45,653	132,080	74,457	108,809	104,579
2021	270,939	223,490	51,238	241,378	86,626	173,205	162,375
2022	591,115	419,955	56,497	482,552	97,725	276,781	251,047
2023	2,043,320	966,314	61,401	1,378,324	108,557	464,309	399,111

## Data Availability

All data generated or analyzed during this study are included in this published article and in the Appendix A.

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
