# Peer review of "Demographic Control Measure Implications of Tuberculosis Infection for Migrant Workers across Taiwan Regions"

_ijerph, 2022, doi:10.3390/ijerph19169899_

Round 1

Reviewer 1 Report

The authors present a mathematical modelling analysis for different packages of interventions to reduce TB burden arising from and to migrant workers in Taiwan. The model is an adaptation to the Jia et al method (2008, ref 23) and is adequately explained and described. Overall, the work is valuable and might be of interest as an academic exercise exploring TB interventions in Taiwan. However, I see that a major limitation of this work is the lack of validation of the model against local data: if we can't ascertain that the model is able to reproduce past trends in Taiwan, how can we make conclusions about its projections? 

Some more specific comments are below:  

Abstract

Try to use synonyms of "control". Also, what is being measured here to speak about effectiveness? Incidence? 

 Introduction:

- Line 44 and 46: "...to those of Taiwanese" , should be "... to that in Taiwanese" as it is the subject is incidence.

- Line 73: This means only one route between infectious locals to a susceptible migrant? (counterintuitive given the background info about TB being imported)

Methods:

- Line 96: The model was not developed for the purpose of this paper. The authors used a previously described model by Jia et al, and adapted it for introducing the interventions.  

- Section 2.2. Table 1: It would be useful to add an equation for BetaM and BetaLM for clarity.

- Line 114: If referring to TB as a microorganism better to say Mycobacterium tuberculosis or just TB, instead of "bacteria"

- Line 114: kL and kM are reactivation rates expressed as (yr^-1), and the text speaks of recent and remote reactivation, but the model has only one route of reactivation with an average rate of 0.004 yr^-1, which is an average rate of 250 years. The latter is clearly a remote route to reactivation. Amend the text to clarify that only remote reactivation is allowed. Also, the model suggests that a  fast route from latent TB to active TB (T) is through re-infection [ q(BmTm+BlmTlm) ]. This should be mentioned.

- The model is well explained and in line with standard TB transmission modelling.

Results:

- Line 184: Are these your results? These are National Statistics on immigration that the authors display in Fig 2 (is this an original figure? or should a source be cited?) If these are truly original results, then the methods should mention how the information was collected, otherwise mention the source.

 - Line 204: Is this incidence rate in the Philippines or among migrant Filipino workers in Taiwan? How is that a finding of yours if it is not part of your analysis? Is this a modelling output or surveillance data?

- Figure 3: Source of data?

- Perhaps Fig. 4 can be replaced with a table?

Discussion

- Line 341-342: WHO targets are set in TB incidence terms which is not used in this manuscript. Results here show "prevalence" of latent (L) and infectious TB (T) but NOT incidence. So, if the statement here is to be supported, the authors need to add modelled incidence reduction estimations for 2025 and 2035. 

- The most important limitation is not mentioned here: This mathematical model was not calibrated (validated) against local data, this is, we don’t know if the current model is able to reproduce past TB trends in Taiwan. 

Reviewer 2 Report

- The authors should explain about the novelty of the research.

-Your manuscript needs a professional revision for the language.

- Add clearly the hypothesis, aims and goals of this work to the last paragraph to your introduction.

Reviewer 3 Report

Thank you for the opportunity to review the manuscript, "Demographic Control Measure Implications of Tuberculosis 2 Infection for Migrant Workers Across Taiwan Regions 3" submitted to the International Journal of Environmental Research and Public Health. The major areas for improvement are specific to the presenting the literature review explaining the theoretical framework or conceptual model, providing the complete description of the a priori model (with a figure indicating the a prior relationships), and providing more specificity about the methods. From my assessment, the current manuscript has mixed information that should be in a backgrounds section with the information in the methods section -assuming the model was developed a prior based upon the literature.

If the above assessment is correct, the study needs to present the framework or model used for the research question in the background section (following the introduction). Even when data sets are used to test a model developed a prior, the model should be clearly presented and justified in the background section. As the authors realize, an a priori model is necessary for a study to test a model rather than the other alternative a data mining activity to develop a model post priori. This is something the authors can correct through a revision process. Furthermore, the manuscript needs to present the methods and procedures in a systematic manner. Again, using data to test an a prior model is not unique in terms of the description of the data and the application to the model for analysis. The discussion section seems appropriate assuming the model was developed a prior. 

STUDY PURPOSE

Although the objective of the study was stated to "assess control measure efforts on reducing TB infection among migrant workers for the optimal infection control in Taiwan regions" there is no specific information about the concepts under investigation. For example, what are the specific control measures, and what is optimal infection control for this study?

INTRODUCTION

-----This information does not provide a clear and concise overview of the theoretical framework of conceptual model used for this study.

A mechanistic, population transmission dynamics-based model can offer a useful tool for analyzing the spread of infectious diseases and evaluate the relative effectiveness of different measures [14]. The mathematical modeling of TB transmission has a long history and being applied in numerous studies with more refined and optimized models constructed [15-23].

-----Where is this model described in relationship to the concepts addressed in this manuscript?

In this study, we adopted a well-established TB dynamic model [23] to examine the migrant-based TB transmission in Taiwan considering transmission routes between infectious local and migrant populations.

-----Where is the evidence of the literature review in the manuscript? What literature was used for the development of the model versus the literature applied for the data analysis?

-----In the context of the next statement, I would expect to see an introduction and then a background section explaining the model. 

We performed a broad literature data analysis to implement model parameterization. Various scenarios were investigated to explore the control scheme for reducing the number of individuals with latent and infectious TB. Three control strategies of social distancing control, early screening, and DOTS and their combinations were considered. Hence, the objective of this study is to assess control measure efforts on reducing TB infection among migrant workers for the optimal infection control in Taiwan regions.

METHODS

The methods section needs to be organized with the following subheadings. These are recommended as testing a model (developed a priori) with secondary data is not an unusual activity. However, the information should be reorganized with normal subheadings. Again, the model seems to be presented in the methods section rather than the background section. The model will need to be referenced for the analysis, and the results will present the outcome of the analysis to the model.

Study Design (with justification and citations)

Setting and Sample

Data Collection

Data Analysis

As one last note, the manner in which the data was cleaned from the original data set for use in this study was not specified. 

RESULTS

-----The results section seems to be mixing discussion material with the results. 

Round 2

Reviewer 1 Report

I thank the authors for addressing my comments. My main concern about the lack of model validation has been highlighted and noted in the discussion which is important. Other amendments have improved the quality of the paper.

Reviewer 3 Report

Thank you for the opportunity to review, Demographic Control Measure Implications of Tuberculosis Infection for Migrant Workers Across Taiwan Regions, submitted to the International Journal of Environmental Research and Public Health. The revisions to the manuscript seem reasonable in response to the review observations. I have one further comment for a minor revision. 

Introduction. This information does not provide a clear and concise overview of the theoretical framework of conceptual model used for this study.

Please add the additional information to a section called the background following the introduction. This is the customary way to provide an "introduction" to the topic before moving on to provide the "background" for the study. As such, I would recommend an introduction section and background section to provide clear "sign posts" due to the density of the information.

Finally, the manuscript could use a professional editing to address grammar, syntax, form, flow, and style. There are many "rough" areas that could be clearer with more editing.
